computer modelling and simulation/energy/environmental engineering

Chinese solar greenhouse, thermal performance, thermal insulation layer, numerical simulation, heat storage

**Authors for correspondence:**
Yiming Li
e-mail: liyiming@syau.edu.cn
Tianlai Li
e-mail: lxa10157@syau.edu.cn

# Effect of external thermal insulation layer on the Chinese solar greenhouse microclimate

Zilong Fan[1,2,4], Xingan Liu[1,2,4], Xiang Yue[3], Lei Zhang[1,2,5], Xiaoyu Xie[6], Yiming Li[1,2,3] and Tianlai Li[1,2,4]

[1]Key Laboratory of Protected Horticulture, Shenyang Agricultural University, Ministry of Education, [2]National and Local Joint Engineering Research Center of Northern Horticultural Facilities Design and Application Technology (Liaoning), [3]College of Engineering, Shenyang Agricultural University, [4]College of Horticulture, Shenyang Agricultural University, and [5]College of Information and Electrical Engineering, Shenyang Agricultural University, No. 120 Dongling Road, Shenhe District, Shenyang 110866, People's Republic of China
[6]Institute of Agriculture and Animal Husbandry Sciences, Tongliao 028000, People's Republic of China

YL, 0000-0001-8764-5589

In order to optimize the heat preservation capacity of Chinese solar greenhouse (CSG) and further reduce energy consumption, we clarified the mechanism of the external thermal insulation layer that affects the microclimate environment of CSG. The most suitable external insulation layer thickness (EILT) of the solar greenhouse envelope structure in high latitude and cold region has been indicated. A three-dimensional mathematical model was developed based on computational fluid dynamics and verified using experimental measurement. The temperature variations, heat variations and economic benefit were analysed. The results indicated that covering the outer surface of the enclosures with a thermal insulation layer could effectively increase the greenhouse temperature by 1.2–4.0°C. The influence degree of the external thermal insulation layer on the greenhouse microclimate was as follows: sidewall (SW) > north wall (NW) > north roof (NR). In high-dimensional and cold areas, covering the outer surface of all enclosures with insulation layer as the suitable solution could raise the greenhouse air temperature maximally. The suitable EILT of each maintenance structure was obtained as follows: NW 80 mm, SW 80 mm, NR 100 mm.

# 1. Introduction

Chinese solar greenhouse (CSG) is an energy-saving facility that allows long growing seasons without active heating in high latitudes under cold climate conditions. Most solar greenhouses can provide a suitable microclimate for growing crops, even producing various vegetables during winter [1]. According to statistics, the applied area of solar greenhouse in China reaches $9.3 \times 10^5$ hm$^2$, and the annual production of vegetables exceeds 100 million tons. Compared with the multi-span greenhouse, it saves 750 tons of coal per hectare [2]. At present, CSG has solved the problem that people have difficulty in eating vegetables in winter, and is designed to achieve high quality and yield of crops. The air temperature in the solar greenhouse must be maintained at a suitable level for a long time to ensure normal crop growth. The indoor environment of a solar greenhouse mostly depends on its materials and structure, critical in storing and releasing heat [3–5]. In recent years, many researchers have done relevant studies on the thermal environment inside the solar greenhouse. They have considered different influencing factors to explore the thermal performance of solar greenhouse [6–11]. The most important reason for the drop in indoor air temperature is internal heat loss. The envelope structure of solar greenhouse has an excellent heat storage effect. However, due to the envelope structures with high heat storage characteristics having high heat transfer coefficient, the heat stored is not conducive to maintenance. Heat absorbed by the envelope structures can easily flow outside. Consequently, the study of external thermal insulation is of great significance to improve the thermal insulation performance of the greenhouse.

Solar greenhouse external insulation is divided into two parts: soft insulation and hard insulation. Soft insulation is to cover the outer surface of the south roof of the solar greenhouse with thermal insulation layer at night, by covering the outer surface of the film with thermal insulation quilt. Large number of studies have explained the effect of insulation parameters of insulation quilt on thermal performance of greenhouse, and the research theory has been quite mature [12,13]. The hard insulation consists of three parts: the north wall (NW), the side wall (SW) and the north roof (NR). At present, large number of studies are focused on the thermal insulation of the NW. The combination of thermal insulation material and wall is studied for the interior of the NW. Zhang *et al.* [11] used numerical simulation to prove that the greenhouses with a clay brick wall (0.6 m thick) insulated with polystyrene boards (0.1 m thick) and a soil wall (3 m thick) preserved more heat than the greenhouse with a hollow concrete block wall (0.6 m thick). Guan *et al.* [14] proposed a three-layer wall with phase change material (PCM) and validated the suitability and correctness of the methods used to construct the three-layer wall. Chen *et al.* [5] developed an active-passive ventilation wall with PCM. A comparative study was carried out using experimental and numerical methods to justify its advantages over conventional walls. Tong *et al.* [15,16] used the computational fluid dynamics (CFD) model to simulate the temperature variations for different wall materials in a CSG. They analysed the dynamic thermal characteristics of 15 wall configurations with one or two insulation layers during clear winter days. On the exterior surface of the NW, Li *et al.* [3] investigated the impact of the heat insulation wall could be weakened by decreasing the heat loss through the greenhouse enclosure.

Although the NR of the greenhouse only accounts for 12.3% of the overall heat dissipation area, it is located on the north windward side. It has a relatively high heat dissipation intensity. The actual heat dissipation coefficient is more than 20% higher than that of the total heat dissipation area [10,17]. Current studies had ignored the importance of insulation for SWs and the NR. Instead, and importantly, all three should be studied in a systematic combination. In greenhouse construction, three elements need to be explicit: material, location and thickness. As an excellent external insulation material with low cost, low density and low heat transfer coefficient, polystyrene board has been widely used in external thermal insulation [3,18–20]. At present, people mainly rely on construction experience to build external insulation. Which exterior surface of the enclosure is the most cost-effective to cover with polystyrene board? In addition, there is no systematic and complete size of external insulation. The insulation performance cannot be maximized under the premise of cost saving.

Therefore, this research filled this gap, and the thermal characteristics of different external placements of insulation layer were explored by combining numerical and experimental research. Comprehensively combining the three elements of hard insulation to study systematically the influence mechanism of external insulation layer on all envelope structures. The external insulation layer thickness (EILT) was further optimized. The rational allocation of external insulation layer based on crop overwintering production in solar greenhouse in high-dimensional and cold areas is put forward. CFD used in this study is an effective simulation tool for predicting the microclimate in solar greenhouses with reliable

results and low costs [21,22]. The external placement of the insulation layer was classified into five types: the greenhouse with no insulation layer on the exterior of the maintenance structures (NO), the greenhouse with insulation layer on the exterior of the NW, the greenhouse with insulation layer on the exterior of the SW, the greenhouse with insulation layer on the exterior of the NR, and the greenhouse completely covered with insulation layer (CC). The results of this study can provide basic theoretical guidance for the rational configuration of the thermal insulation layer in CSG production practice.

# 2. Material and methods

## 2.1. Experimental arrangement

The experimental CSG, Liao-Shen type CSG, was located in Shenyang Agriculture University, Liaoning Province, China (41.8° N, 123.4° E). The climate is a typical temperate continental monsoon climate. The lowest temperature is −20°C, and the highest temperature is 31°C in a whole year. The greenhouse was 60 m long with an east–west orientation. The span and the height of the greenhouse were 8.0 m and 4 m. The height and thickness of the NW were 2.7 m and 0.37 m, respectively. The outer surface of the NW was covered with a 0.11 mm thick polystyrene board. The NR was a waterproof insulation layer composed of wooden boards and straw felt wrapped in a waterproof polyolefin film. The south roof was covered with a 0.11 mm plastic film and covered with a 0.04 m thick cotton blanket at night for insulation. The greenhouse was sealed to avoid the effects of airflow. The experiments were conducted on 23 January 2020, which represented severe winter. Figure 1a shows the layout of the CSG and schematic for energy transfer processes in a greenhouse. Solar radiation entering the greenhouse will flow outside through the NW, south roof, SW and NR. Figure 1b shows the schematic diagram of the experimental greenhouse.

To determine the thermal environment inside the CSG, a Decagon micro-meteorological monitoring system was installed outdoors at a distance of 1 m from the greenhouse. The instrument had a temperature measurement accuracy of ±0.7°C. The Em50/R/G data collector was used as the core system, and sensors measuring different environmental parameters were integrated into the same collector for data collection. It was set to record data every 10 min. Six monitoring points were placed inside the CSG to obtain quantitative data on internal air temperature, and the TR-71UI temperature recorder was used to record data every 10 min. All sensors were installed in the middle cross-section of the greenhouse. Environmental data recorders automatically collected data during measurement [23]. To determine the temperature distribution of the NW, the FLUKE 2638A data collector was adopted to measure the temperature of each layer of the NW in the CSG. Data were recorded every 10 min. Figure 2 shows the location distribution of the measurement points.

## 2.2. Mathematical model

The material composition of the model included: the indoor air was an incompressible ideal gas, the NW and SWs were mainly composed of clay brick walls, and the NR was mainly composed of wood panels. The south roof used the polyolefin film for daylighting and heat absorption during the day. The greenhouse was covered with the insulation quilts at night. Polystyrene board was also included in the calculation for greenhouses as external insulation to insulate the external surfaces of the different envelopes. Table 1 shows the thermo-physical properties of the materials in the experimental CSG. A physical model was constructed for experimental verification with the same physical parameters as the greenhouse used for experimental data collection. The comparison model of the greenhouse showed no polystyrene board on the outer surface of maintenance structures. In the other four cases, the polystyrene board was respectively covered on the outer surface of the NW, NR, SWs and full coverage. In addition, greenhouses were modelled with different envelope outer insulation thicknesses established and calculated by the validated model building method. Figure 3 shows the schematic diagram of the structure model for the five types of external placement of polystyrene board. To accurately calculate the effect of external insulation on the greenhouse thermal environment, changes in humidity, transpiration of plants and ventilation conditions inside the greenhouse were not considered. Most of the heat in the greenhouse dissipated to the external environment through the enclosures.

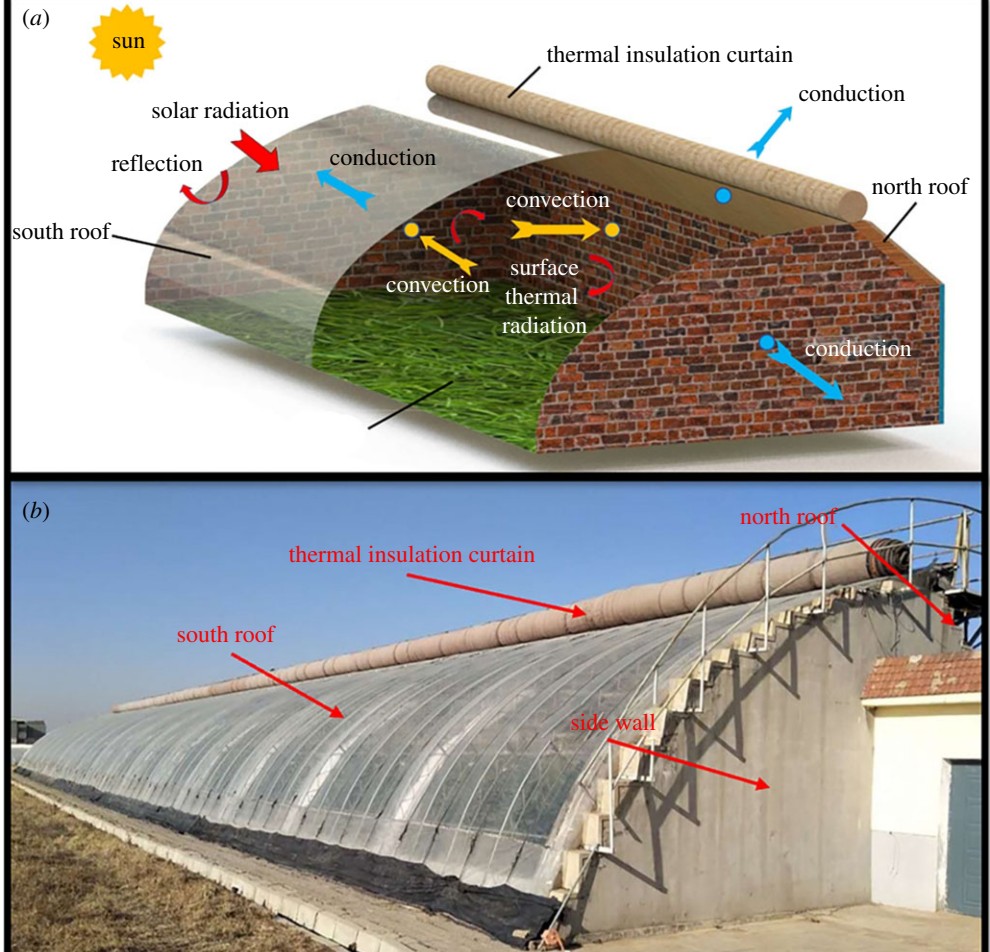

**Figure 1.** (*a*) Schematic for energy transfer processes in a greenhouse; (*b*) schematic diagram of the experimental greenhouse.

The principle of CFD technology is to solve the differential equation controlling the fluid flow numerically and obtain the discrete distribution of the flow field in the continuous region [24], including three fundamental conservation equations: mass, momentum and energy. The conservation of mass partial differential equation (equation (2.1)) means that the net mass flow into the control body is equal to the increase in mass per unit time due to the change in density in the control body [25–27].

$$\frac{\partial \rho}{\partial t} + \nabla \cdot (\rho \boldsymbol{V}) = 0, \tag{2.1}$$

where $\rho$ is the air density, $t$ is the time, $\boldsymbol{V}$ is the velocity vector.

Fluid mechanics belong to the classical category of mechanics, so the application of momentum conservation equation is essentially Newton's second law. The momentum partial differential equation (equation (2.2)) can be described as the momentum increment per unit time is equal to the net momentum flow into the system plus the force on the system.

$$\frac{D\boldsymbol{V}}{Dt} = f_{\mathrm{b}} - \frac{1}{\rho}\nabla p + \frac{\mu}{\rho}\nabla^2 \boldsymbol{V} + \frac{1}{3}\frac{\mu}{\rho}\nabla(\nabla \cdot \boldsymbol{V}), \tag{2.2}$$

where $f_{\mathrm{b}}$ is the volume force item, $p$ is the pressure, $\mu$ is the dynamic viscosity.

The energy partial differential equation (equation (2.3)) can be described as the rate of change of the kinetic energy and internal energy of the material in the region is equal to the work done by the volume force and the area force per unit time plus the heat given to the material per unit time from the outside.

$$\rho \frac{D}{Dt}\left(\hat{u} + \frac{V^2}{2}\right) = \rho \cdot \boldsymbol{V} + \nabla \cdot (\boldsymbol{V}\tau_{\mathrm{ij}}) + \nabla(\lambda \nabla T) + \rho q, \tag{2.3}$$

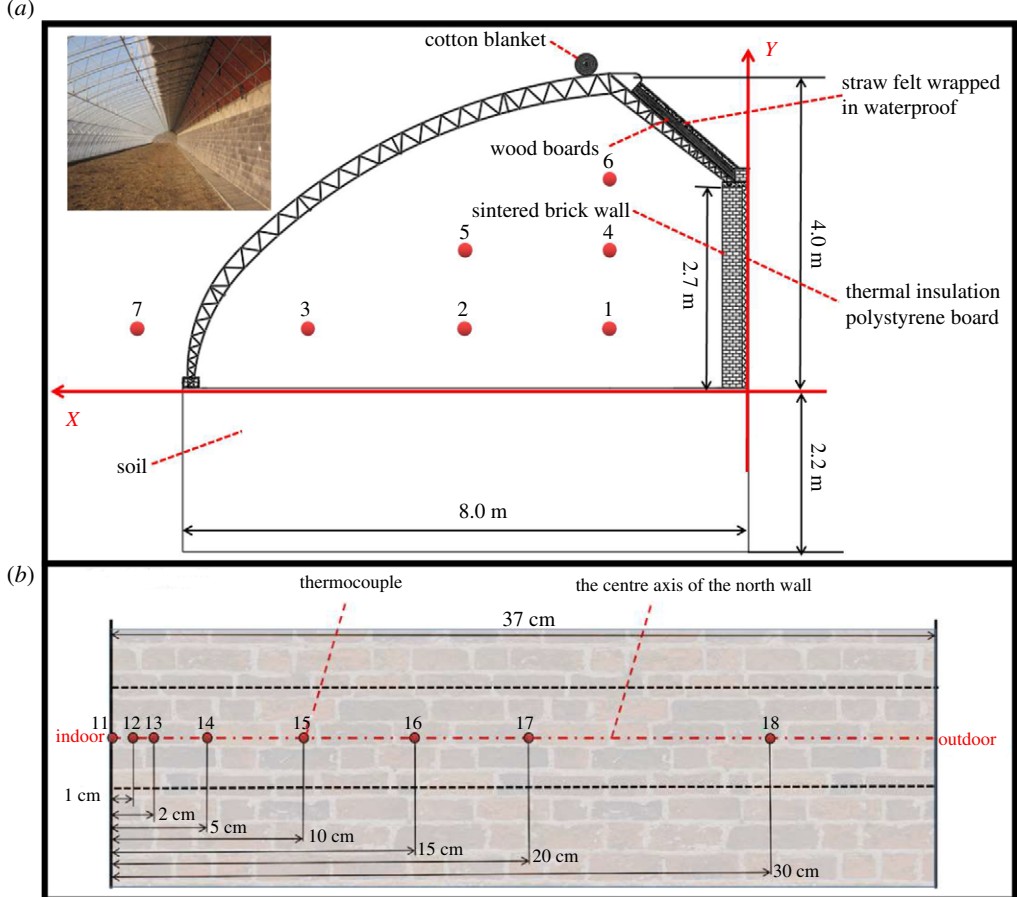

**Figure 2.** A sectional view of the measurement points in the centre: (*a*) greenhouse interior air; (*b*) the north wall.

**Table 1.** Thermo-physical properties of materials used in simulation.

| material | density (kg m$^{-3}$) | specific heat capacity (J kg$^{-1}$ °C$^{-1}$) | thermal conductivity (W m$^{-1}$ °C$^{-1}$) |
|---|---|---|---|
| internal air | ideal-gas | 1006.43 | 0.024 |
| clay brick | 1600 | 1051.1 | 0.5 |
| polystyrene board | 30 | 2414.8 | 0.041 |
| wood board | 550 | 2510 | 0.29 |
| polyolefin film | 950 | 1600 | 0.19 |
| cotton blanket | 150 | 1880 | 0.06 |
| soil | 1700 | 1010 | 0.85 |

where $\hat{u}$ is the internal energy of air per unit mass, $V$ is the velocity of air per unit mass, $\tau_{ij}$ is the effective viscosity shear, $\lambda$ is the thermal conductivity, $T$ is the temperature, $q$ is the unit mass of air receives heat from the outside through radiation.

In order to improve the convergence rate of the governing equation, the indirect flow viscosity of fluid molecules is ignored in the standard $k-\varepsilon$ model (equations (2.4) and (2.5)) [28,29].

$$\frac{\partial}{\partial t}(\rho k) + \frac{\partial}{\partial x_i}(\rho k u_i) = \frac{\partial}{\partial x_j}\left[\left(\mu + \frac{\mu_t}{\delta_k}\right)\frac{\partial k}{\partial x_j}\right] + G_k + G_b - \rho\varepsilon - Y_M + S_k \qquad (2.4)$$

and

$$\frac{\partial}{\partial t}(\rho\varepsilon) + \frac{\partial}{\partial x_i}(\rho\varepsilon u_i) = \frac{\partial}{\partial x_j}\left[\left(\mu + \frac{\mu_t}{\delta_\varepsilon}\right)\frac{\partial\varepsilon}{\partial x_j}\right] + C_{1\varepsilon}\frac{\varepsilon}{k}(G_k + G_{3\varepsilon}G_b) - C_{2\varepsilon}\rho\frac{\varepsilon^2}{k} + S_\varepsilon, \qquad (2.5)$$

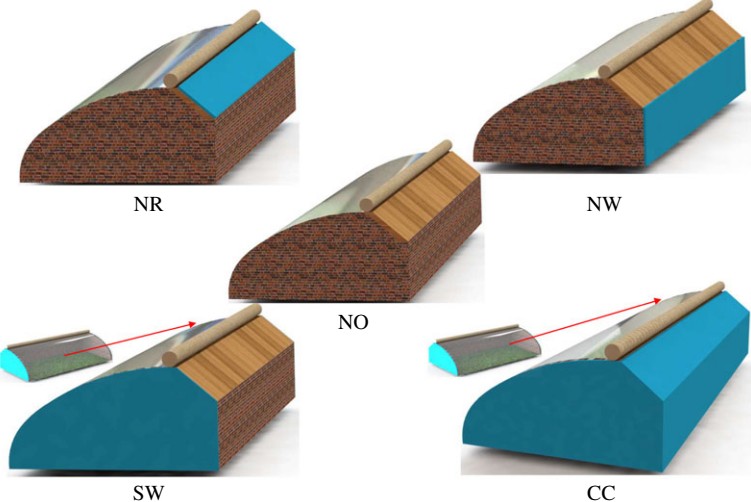

**Figure 3.** Schematic diagram of the structure model for the five types of external placement of polystyrene board.

where $G_k$ represents the generation of turbulence kinetic energy due to the mean velocity gradients. $G_b$ is the generation of turbulence kinetic energy due to buoyancy. $Y_M$ represents the contribution of the fluctuating dilatation in compressible turbulence to the overall dissipation rate. $C_{1\varepsilon}$, $C_{2\varepsilon}$ and $C_{3\varepsilon}$ are constants. $\delta_k$ and $\delta_\varepsilon$ stand for the turbulent Prandtl numbers for $k$ and $\varepsilon$, respectively. $S_k$ and $S_\varepsilon$ refer to user-defined source terms. The turbulent viscosity, $\mu_t$ is computed by combining $k$ and $\varepsilon$ as follows:

$$\mu_t = \rho C_\mu \frac{k^2}{\varepsilon}, \tag{2.6}$$

where $C_\mu$ is constant.

The $P-1$ radiation model is suitable for cases with optical thickness greater than 1. It can be used to consider the scattering or absorption effects of particles in other forms in the gas medium, with high computational efficiency.

$$q_r = -\frac{1}{3(a + \delta_s) - C\delta_s}\nabla G, \tag{2.7}$$

where $a$ is the absorption coefficient, $\delta_s$ represents the scattering coefficient, $G$ is the incident radiation and $C$ refers to the linear-anisotropic phase function coefficient.

## 2.3. Numerical details

Figure 4$a$ shows the model zone containing the target greenhouse geometry and the grid of the domain. The average temperature in the greenhouse was monitored and its distribution was analysed. As can be seen in figure 4$b$, the cells number was refined from $7 \times 10^5$ to $1.9 \times 10^6$. The grid number was divided into five gradients. The average air temperature fluctuation between the third (about $1.3 \times 10^6$ cells) and the fourth level ($1.6 \times 10^6$ cells) of the mesh refinement was not significant. The accuracy of the results did not improve significantly when the number of meshes increased to $1.9 \times 10^6$. Therefore, in order to reduce the computational burden and ensure the accuracy of the simulation calculation, the same meshing method was used for all cases in this study. The total number of grids was maintained at $1.3 \times 10^6$. The hexahedral mesh was adopted, and the local meshes were refined to improve the calculation accuracy when meshing the model [30]. The mesh skewness was below 0.6. To assess the time-step independency of the results, the computed average greenhouse air temperature in a period of 10 s was monitored while using three different time steps consisting of 1, 5 and 10 s. Figure 4$c$ shows that the computed temperature applying in this wide range of time steps (from 1 to 10 s) is almost the same. Considering the computational costs, the maximum acceptable time step of 10 s was selected for this study.

The boundary and initial conditions were determined based on the measured indoor and outdoor temperatures. The transient three-dimensional turbulence model, standard k-ε model, and standard wall function were applied to the wall surface [31]. The semi-implicit method for pressure linked

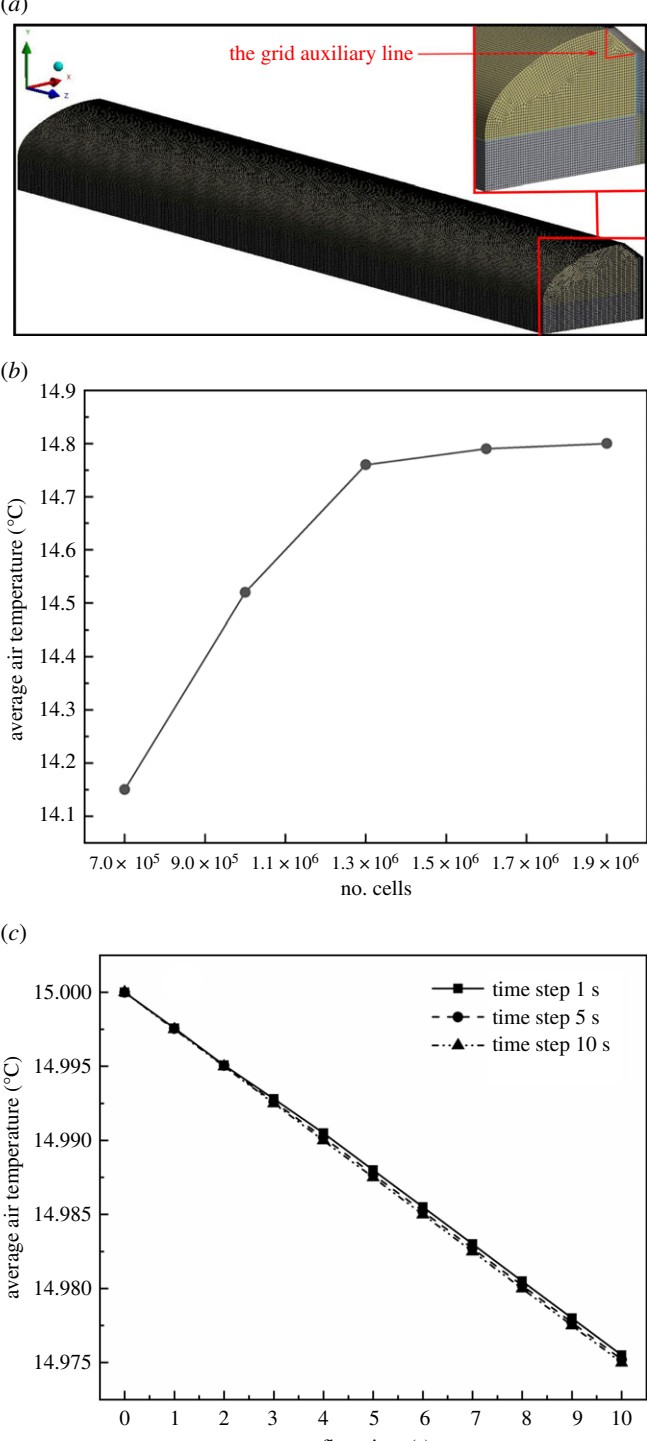

**Figure 4.** (*a*) The model zone including the target greenhouse geometry and the grid of the domain; (*b*) computed average air temperature based on different grid of the domain; (*c*) computed average air temperature using three different time steps of 1, 5 and 10 s.

equations (SIMPLE) scheme was employed based on the pressure and velocity coupling solver [32]. Second-order upwind was adopted for density and energy. The convergence accuracy of all equations was configured as $10^{-3}$, while the convergence accuracy of energy equation and radiation equation was configured as $10^{-6}$ [33]. After the residual value was not reduced (for the total residual of all variables in each equation), the iterative step size was finally determined to maximally iterate 20 times for each time-step size.

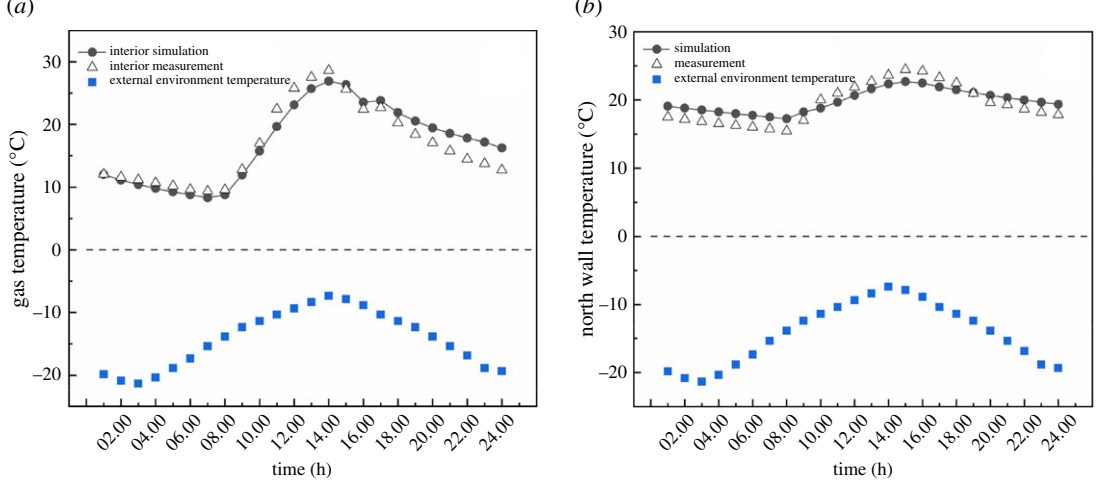

**Figure 5.** The average temperature of internal air (*a*) and north wall (*b*) obtained by numerical simulation in comparison with experimental measurement.

# 3. Results

## 3.1. Validation of the numerical simulation

According to the geometric size and material performance parameters of the experimental greenhouse, a simulation model of the actual greenhouse was built, and the simulation calculation was carried out. The predicted results were compared with the experimental results to verify the correctness of the CFD simulations. Figure 5*a* shows that the simulated results coincided with the experimental measurements. The average difference between the experiment and simulation was 0.22°C. The average absolute discrepancy was 9.5%. During the day, the numerical results were slightly lower than the experimental measurements, indicating that the actual greenhouse gained more heat than the mathematical model. However, the predicted results were higher than the experimental measurements after the NR was covered with a blanket quilt in the afternoon, indicating that the actual insulation effect of the thermal blanket quilt was incompetent.

The NW of the enclosures had a large effective volume, and therefore its thermal storage and dissipation performance should be excellent. Figure 5*b* shows that the average temperature difference between the predicted results and the experimental measurements was 0.37°C, with an average absolute error was 7.8%. From 10.00 to 19.00, the experimental data were higher than the numerical results, indicating that more heat was accumulated on the inner surface of the NW in the actual greenhouse. The results show that the model simulates a reasonable indoor temperature environment and validates the accuracy of the numerical simulation. The same method was used to set and solve all the models, and the influence of external insulation layer on CSG microclimate was accurately predicted.

## 3.2. Analysis of the priority of external insulation

The most effective indicator to test the insulation performance of greenhouse is the indoor air temperature. Especially on cold nights, the higher the average night-time temperature, the more effective the insulation is in blocking the heat. Figure 6 shows the average air temperature distributions for the five cases. When the insulation layer was covered on the outer surface of the enclosures completely, the greenhouse air gained the highest temperature throughout the day, with an average night-time temperature increase of 4.09°C. The temperature difference was most remarkable from 14.00 to 0.00. The average temperature of NW was 0.85°C and 0.65°C higher than that of NR and SW, respectively. The average temperature of CC at night was 1.7°C higher than that of NW. The influence of the insulation layer on SW was insignificant due to the small effective volume of the SWs. Moreover, the wood boards in the NR had no heat storage capacity. As a result, the influence of the insulation layer on NR was minimal. The average temperature of NO was the lowest, and the maximum difference of the air temperature at night was 0.6°C.

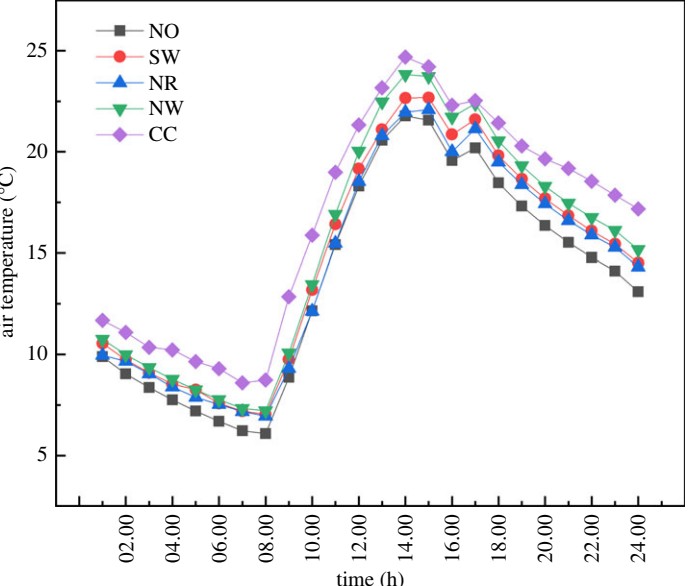

**Figure 6.** Internal air temperature variations for the five types of external placement of insulation layer.

The cost of insulation covering the outer surface of different enclosures is different, and this directly determines the cost of greenhouse construction. For the popularization of heliostats, it is necessary to optimize the construction costs for different locations. According to the actual production experience, the thickness of the outer insulation layer of the NW, SW and NR is 110 mm. To better analyse the economic benefits of different cases, the unit cost was used as the evaluation index. The economic evaluation index was calculated as

$$W_i = \frac{S_i H_i E_i}{T_i - T_0},$$  (3.1)

where $W_i$ is the unit cost; $S_i$ is the coverage area of the insulation layer; $H_i$ is the thickness of the insulation layer; $E_i$ is the unit price of the insulation layer; $T_i$ is the greenhouse air temperature of SW, NR, NW and CC at 0.00; and $T_0$ is the greenhouse air temperature of NO at 0.00.

The maximum temperature difference of the solar greenhouse throughout the day also reflects the strengths and weaknesses of the insulation effect, and the cost of the insulation layer that causes significant differences in the insulation effect is not the same. The combined evaluation of maximum temperature difference and cost can intuitively analyse the priority of covering the outer surface of the enclosure with the insulation layer. Figure 7 shows the temperature difference of internal air with NO as contrast at 0.00 as well as the economy index of the insulation layer. In this study, the unit price of the insulation layer was USD 71.4 m$^{-3}$. Its economic index was the cost of the insulation layer when the internal air temperature was raised by 1°C. The most significant warming effect was achieved by the CC, which raised the temperature by 6°C. The internal air temperature of NW increased by 2.06°C, while the lowest temperature increase of NR was 1.21°C. When the outer surface of the enclosures was completely covered with the insulation layer, the heat loss was prevented. This improved the heat storage–release performance of the greenhouse.

When the insulation layer was singly placed on the outer surface of one side of the enclosures, NW had the best thermal insulation and storage capacity. The unit costs of NW and CC were similar at USD 618.49 °C$^{-1}$ and USD 643.99 °C$^{-1}$, respectively. NR had the highest unit cost (USD 787.99 °C$^{-1}$), but it had the lowest increase in greenhouse air temperature. The economic costs of the insulation layer were not considered; CC as the suitable solution could raise the greenhouse air temperature maximally. SW had the lowest unit cost (USD 282.52 °C$^{-1}$), but it limited the increase in greenhouse air temperature. The influence degree of the external thermal insulation layer on the greenhouse microclimate is as follows: SW > NW > NR.

## 3.3 Thermal performance and optimization of external insulation layer thickness

However, in order to meet the requirements of vegetable production, solar greenhouses in high-dimensional cold regions all adopt CC to withstand the cold external environment. Obviously, it is

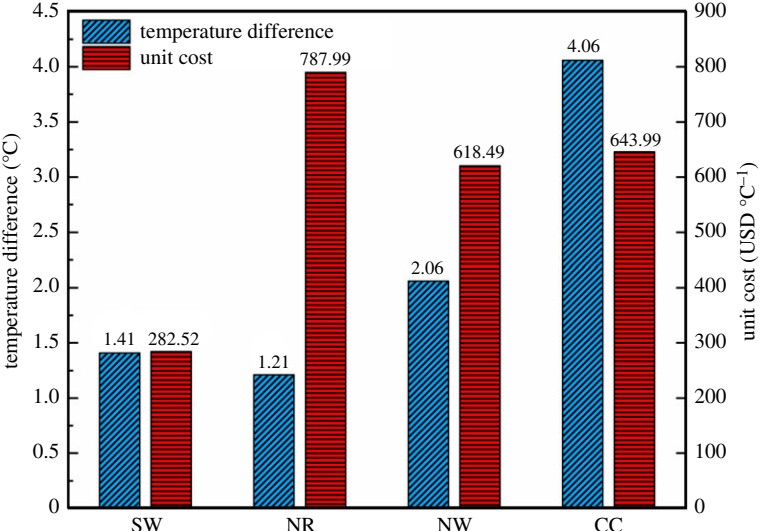

**Figure 7.** The temperature difference of internal air with NO as contrast at 0.00 and the unit cost of four cases.

particularly important to optimize the EILTs for different maintenance structures of CC. In order to reduce the construction cost of the greenhouse and to ensure its excellent thermal insulation performance, the effect law of the EILT on the solar greenhouse maintenance structure was studied in detail in this paper. The temperature distribution visualizes the temperature variation of the greenhouse component structures and better analyses the thermal behaviour of each enclosure. Figures 8–10 describe the effect of different EILTs of the NW, SW and NR on the solar greenhouse microclimate at different typical times. Figure 8a shows that enhancing the EILT will significantly increase the air temperature in the greenhouse, especially in the range of 40–80 mm. Covering the NW with insulation layer also increases the temperature of the NW, but has little effect on the NR and soil. When the EILT is increased to 80 mm or above, the indoor air temperature fluctuation is not significant. Figure 8b shows the temperature distribution of the NW. It can be seen that the effective heat storage volume of the NW increased significantly with increasing EILT. At night, the maximum thickness of the NW could be increased by 120 mm, but the thermal behaviour of the NW tended to be stable when the EILT reaches 80 mm. The above results indicated that the EILT of the NW was more than 80 mm, which could effectively ensure that the NW could store the heat absorbed in the daytime.

As can be seen in figure 9a, the indoor air temperature did not change significantly as the EILT outside the SW increased. Because the SWs were small in building volume and were distributed on the east and west sides of the greenhouse, they had little exposure to the indoor air. Although the SW and the NW had the same thermal parameters, the influence of the SW on air temperature was less than that of the NW due to the difference in volume and location. The effect of EILT on the temperature of the SW was depicted in figure 9b. The effects of EILT on SWs were much greater than indoor air. With the increase of EILT, the effective heat storage volume of the SW increased significantly, and the SW temperature gradually became stable until 100 mm. The maximum thickness of the effective heat storage layer on the SW could be increased by 300 mm.

As can be seen from figure 10a, the increase of thickness had little influence on the internal microclimate. The NR had a low thermal conductivity and good thermal insulation performance, so it lost less heat to the outside world. The NR had limited heat storage capacity and the heat storage effect was poor. Figure 10b showed that with the increase of EILT, the temperature on the NR near the NW increased significantly at night. Lower EILT resulted in more stratified temperature gradients within the NR.

In order to quantitatively analyse the action law of insulation thickness and compare the indoor air temperature at different thicknesses, figure 11 illustrates the comparison of greenhouse air temperatures under EILT with different maintenance structures. Figure 11a showed that the indoor air temperature rose rapidly with the increase of solar radiation in the daytime. At 16.00, the heat loss from the south roof was greatly reduced by covering it with the heat preservation quilt. At the same time, the maintenance structure and the soil had begun to release heat into the interior to counteract the heat loss from the greenhouse, resulting in a slight increase in air temperature. Subsequently, due to the decrease of

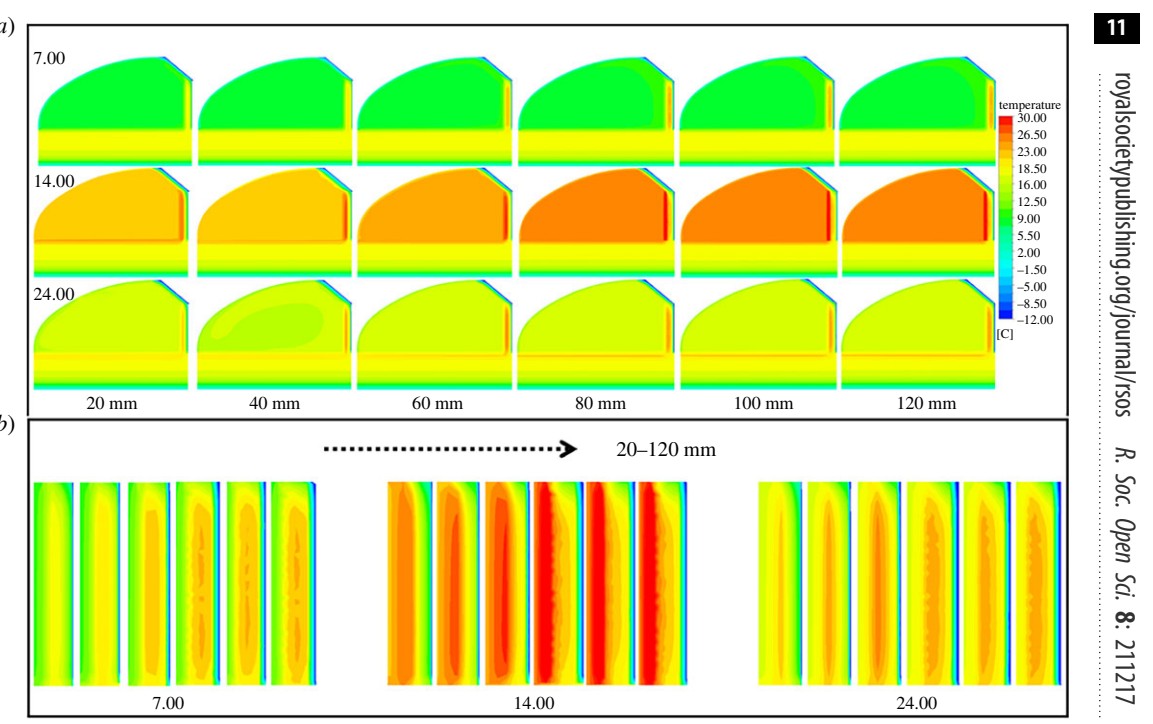

**Figure 8.** The centre section (*a*) and the local north wall (*b*) temperature distribution of solar greenhouse with different north wall EILTs in different typical periods.

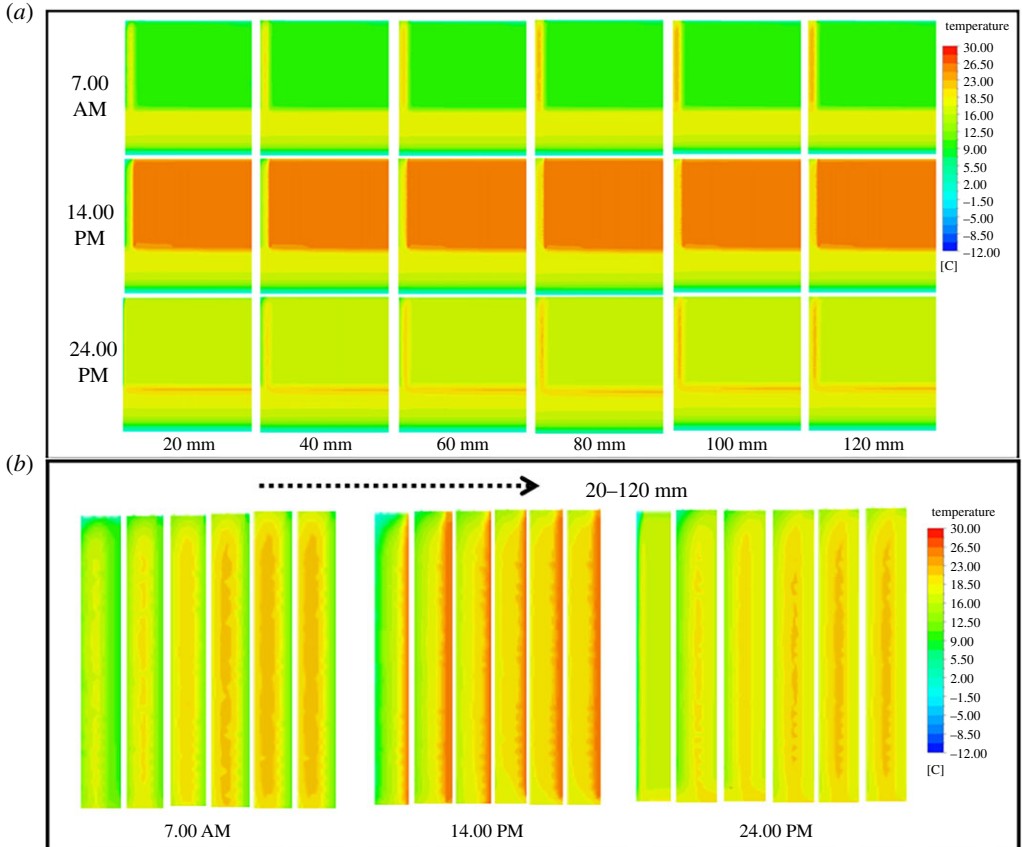

**Figure 9.** The centre section (*a*) and the local side wall (*b*) temperature distribution of solar greenhouse with different side wall EILTs in different typical periods.

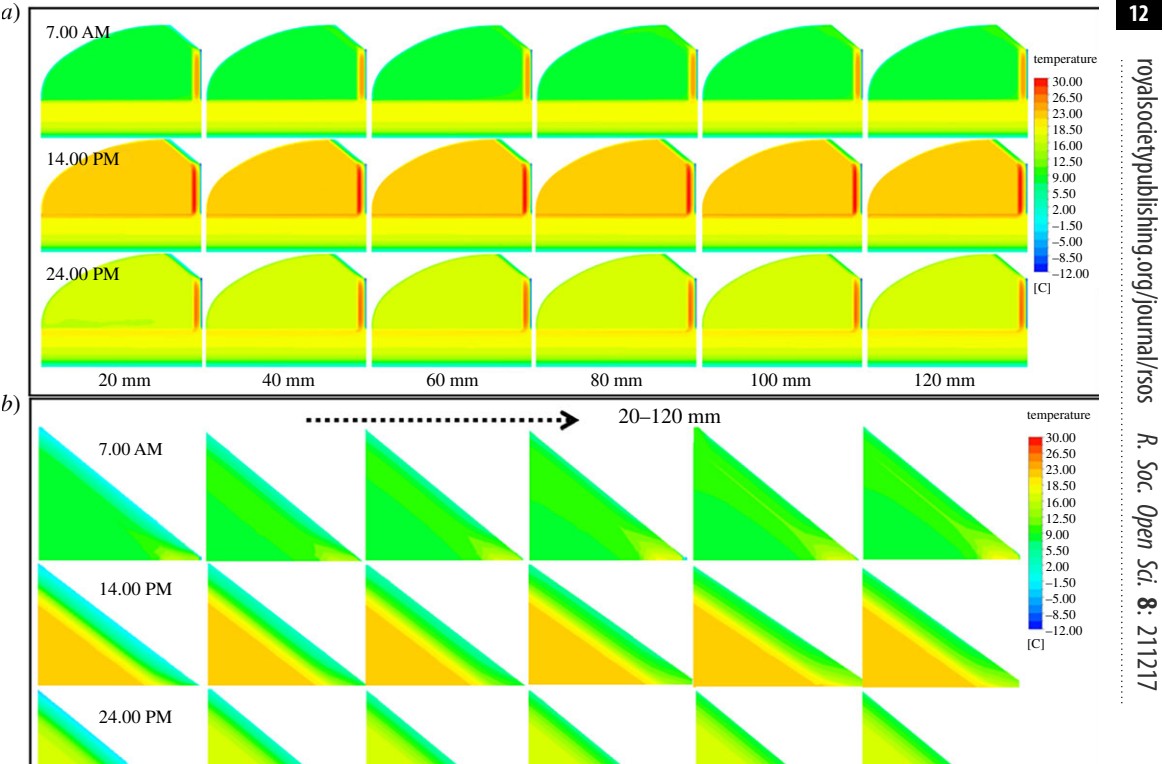

**Figure 10.** The centre section (*a*) and the local north roof (*b*) temperature distribution of solar greenhouse with different north roof EILTs in different typical periods.

outdoor temperature, the convection between inside and outside the greenhouse was enhanced. The heat loss from the maintenance structure increased gradually and the indoor temperature dropped significantly. At noon, the air temperature under different insulation layer thicknesses was significantly different, with a difference of 0.5–4°C. The night-time temperature difference decreased, with an average difference of 0.2–0.3°C for each 20 mm increase in EILT. The maximum temperature difference at night could reach 1.5°C. As can be seen figure 11*b*, the increase of EILT on the SW had little effect on air temperature in the daytime, and the air temperature gradually increased with the increase of EILT at night. The average maximum temperature difference at night reached 0.65°C. It explained that the SW insulation layer mainly affected the air temperature at night. Figure 11*c* showed the effect of EILT on air temperature in the NR. On the whole, EILT on the NR had little effect on air temperature. The increased EILT was less responsive to air temperature, with a maximum air temperature difference of 0.3°C throughout the day.

Air temperature fluctuations are influenced by the temperature of the envelope, which also reflects the phenomenon of heat retention. The thickness of the insulation can be further optimized based on the temperature distribution of the envelope. Figure 12 depicts the effect of EILT on the temperature of different maintenance structures throughout the day. All results showed that the maintenance structure's temperature increased significantly throughout the day when EILT increased in the range before 80 mm. The temperature variation of the NW throughout the day was generally consistent with the trend of air temperature, and the temperature distribution was sinusoidal. When the EILT of the NW exceeded 60 mm, the outdoor environment significantly reduces the NW temperature. The maximum temperature difference on the NW throughout the day was 6°C. Figure 12*b* and *c* showed that the average temperature difference of the maintenance structure was 2°C for EILT of 20–60 mm. When the EILT of the SW reached 100 mm, the influence of external environmental factors on the SW decreased significantly. When the EILT of the NR exceeded 80 mm, the temperature tended to be stable.

The visualization of the average temperature variation of the depth layer of the envelope with time can better analyse the influence law of the insulation thickness on the envelope. The insulation thickness of the NW and SWs no longer responds significantly after 80 mm, while the NR needs to exceed 40 mm. Figure 13*a*

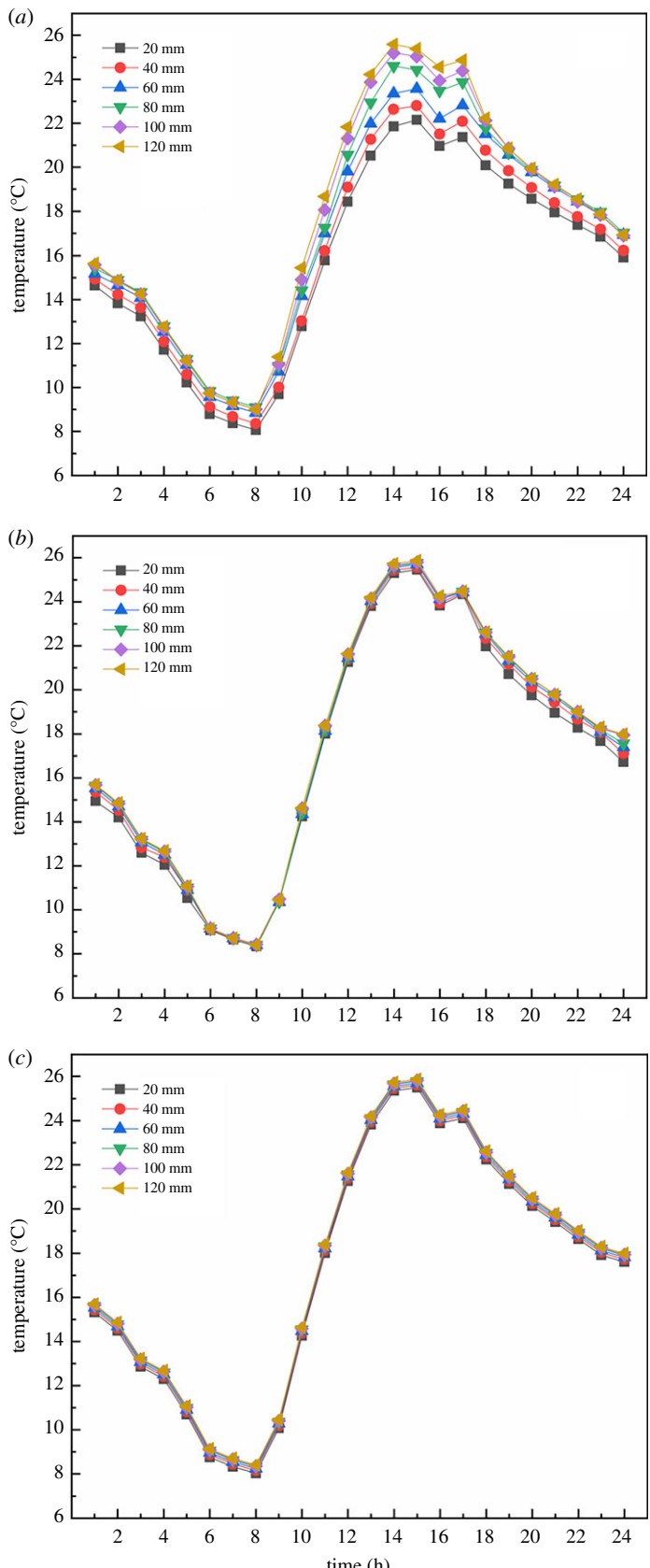

**Figure 11.** The comparison of greenhouse air temperatures under EILT with different maintenance structures: (*a*) north wall; (*b*) side wall; (*c*) north roof.

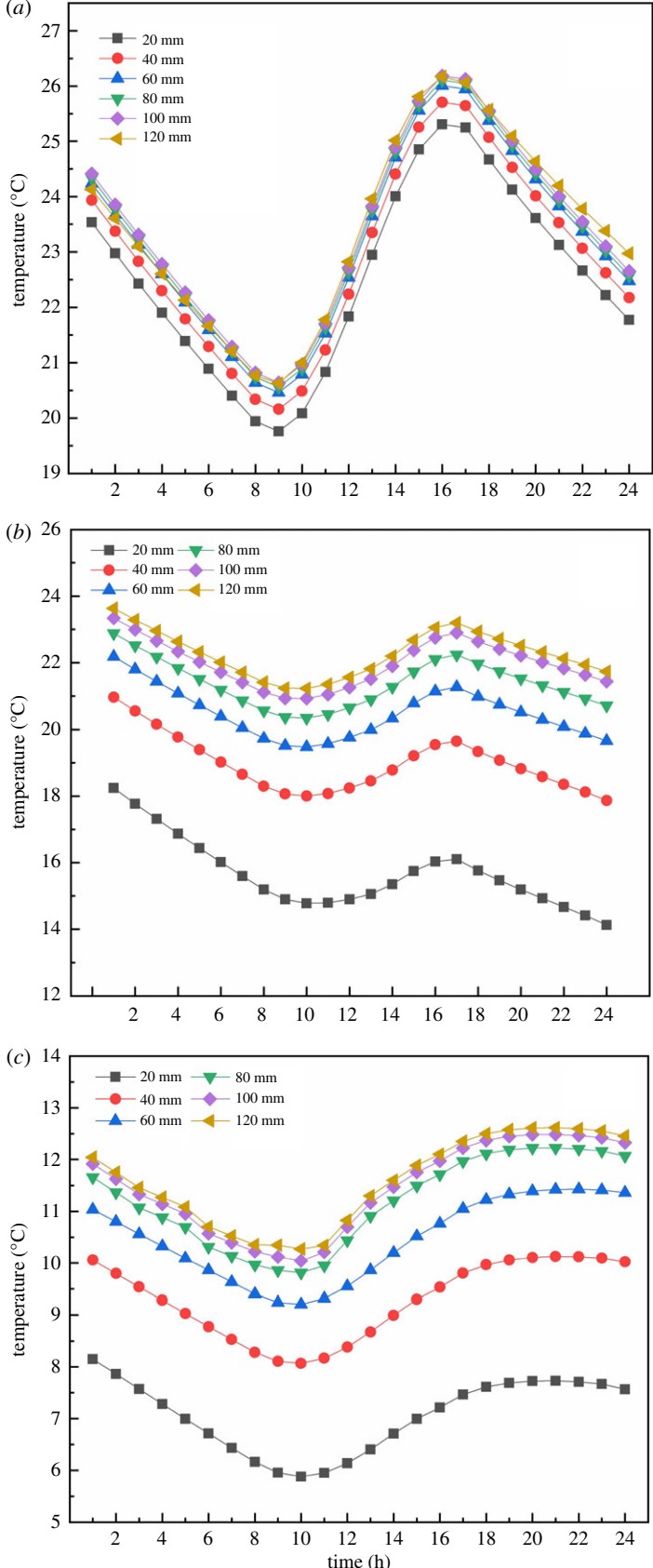

**Figure 12.** The effect of EILT on the temperature of different maintenance structures throughout the day: (*a*) north wall; (*b*) side wall; (*c*) north roof.

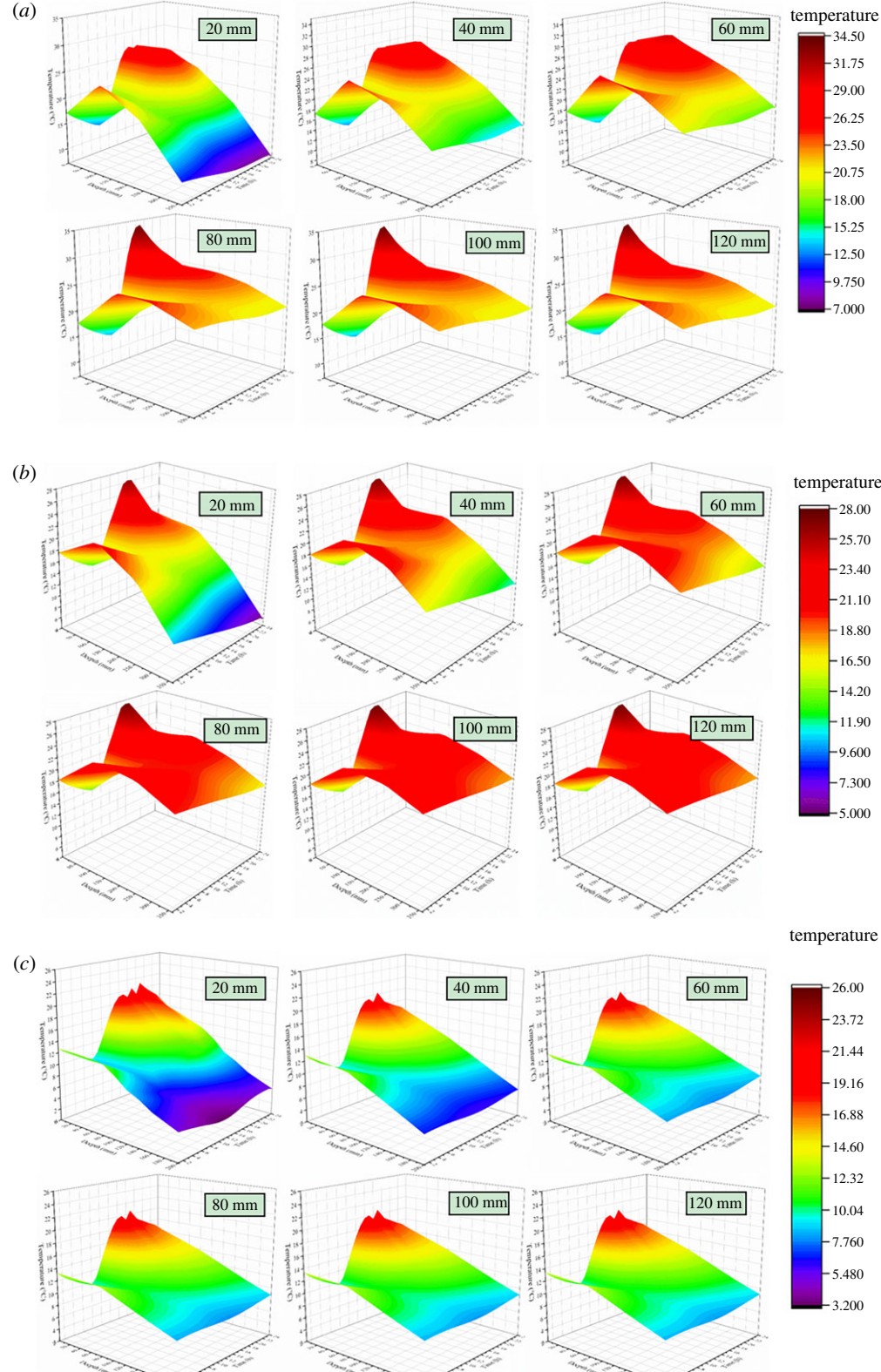

**Figure 13.** The variation of depth layer temperature with time in different enclosure structures with different external insulation layers: (*a*) north wall; (*b*) side wall; (*c*) north roof.

showed that when EILT was 20 mm, very low temperatures occurred in the part of the NW near the outside air. When EILT was between 20 and 60 mm, the internal temperature of the NW 200 and 350 mm away from the inner surface of the greenhouse was seriously affected. The maximum temperature of the NW at noon was 30°C. When EILT exceeded 80 mm, the maximum temperature of the NW at noon could increase to

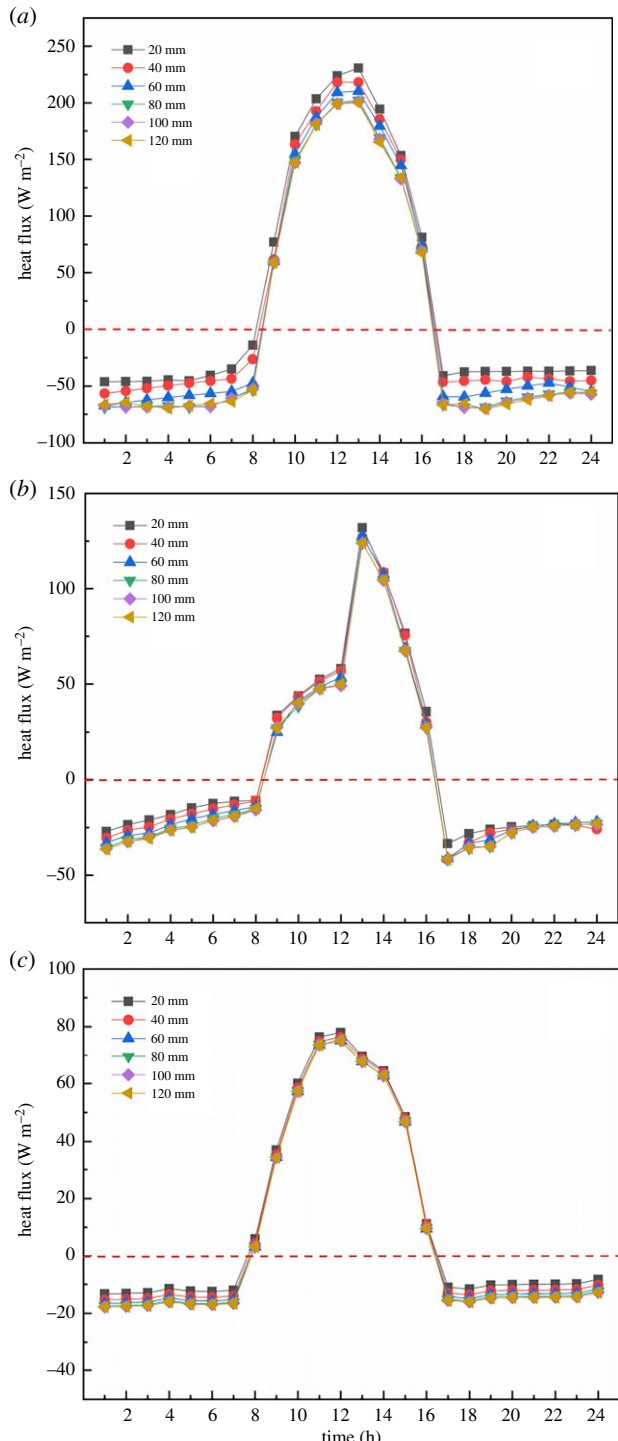

**Figure 14.** The heat flux variation of different maintenance structures at different EILT throughout the day: (*a*) north wall; (*b*) side wall; (*c*) north roof.

35°C. The further away from the inner surface of the NW, the less pronounced the temperature fluctuation. Figure 13*b* showed that the SW temperature near the outdoor air fluctuated sharply when the EILT of SW was less than 40 mm. The high noon temperature did not fluctuate much. EILT of 80–120 mm resulted in inconspicuous SW temperature fluctuations. When EILT on the NR was 20–40 mm in figure 13*c*, the temperature changed significantly in the area near the outside air on the NR. EILT over 40 mm caused a small temperature fluctuation on the NR.

Changes in indoor heat cause temperature fluctuations and heat is lost to the outside through the envelope. Analysis of the heat flow distribution of the envelope can also obtain the law of the action of the thickness of the insulation. Figure 14 shows the heat flux variation of different maintenance

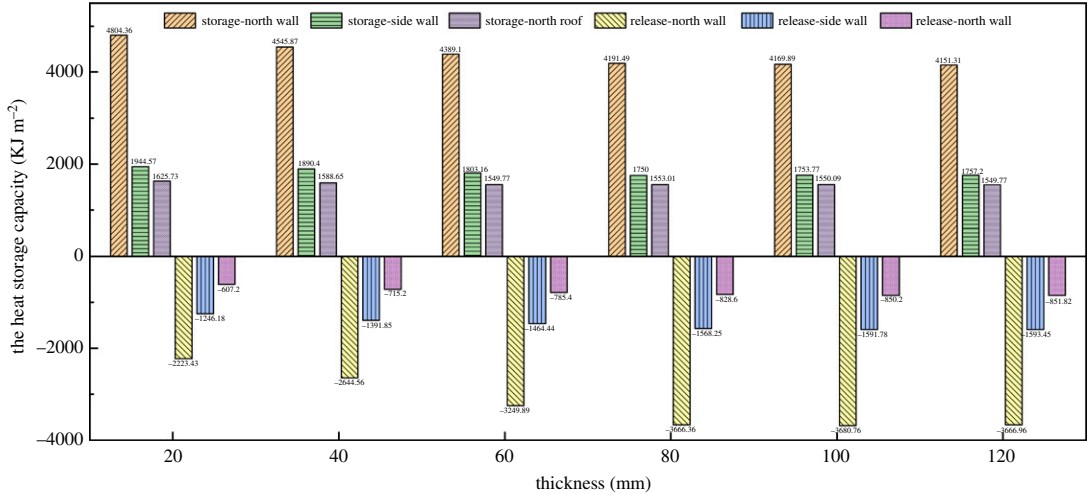

**Figure 15.** The heat storage–release capacity of the maintenance structure covered with external insulation layers of different thicknesses.

structures at different EILT throughout the day. As can be seen in figure 14a, the daytime heat absorption from the greenhouse by the NW decreased with the increase of EILT. The thinner EILT caused a very high temperature difference between indoor and outdoor, creating a heat transfer effect that made the NW more inclined to heat transfer to the outside. The higher the overall thermal conductivity of the NW, the more heat the NW will release to the outside through its exterior. The NW needed to absorb more heat from the greenhouse to keep the NW temperature constant. An increase in EILT caused the NW to release more heat into the greenhouse at night. Figure 14b showed that with the increase of EILT, the heat flux of the SW did not change much in the daytime, while the heat release of the SW increased as well as the NW at night. In particular, because the SW of the analysis was located on the east side of the greenhouse, the amount of solar radiation received by the inner surface of the SW when the sun rose from the east in the morning was relatively limited and the heat flux was relatively low. A sufficiently high solar altitude angle after 12.00 caused a sharp increase in heat flux on the inner surface of the SW. Figure 14c showed that with the increase of EILT in the NR, the change range of heat flux in the NR at night was small, with an average of $-17$ to $-19\,\mathrm{W\,m^{-2}}$.

The heat storage and release capacity of the envelope is quantified by counting the amount of heat storage and release throughout the day. The night-time heat release of the NW can reach 3666 KJ m$^{-2}$ with sufficient insulation thickness, while the SWs and the NR also reach 1590 and 850 KJ respectively. Figure 15 shows the heat storage–release capacity of the maintenance structure covered with external insulation layers of different thicknesses. The heat storage–release capacity index was defined as the heat absorbed and released from the indoor air by the unit area of the maintenance structure throughout the day. Figure 15 showed when EILT increased from 20 to 80 mm, the heat absorbed by the NW decreased and the heat released at night significantly increased. However, when EILT exceeded 80 mm, the thermal change of the NW tended to be stable. The SW and the NW had the similar change of heat storage. The heat storage did not increase when the EILT reached 80 mm, but at night the heat release became stable only when the EILT reached 100 mm. The heat absorption tended to be stable on the NR when EILT reached 60 mm, but heat release reached its maximum and tended to be stable when EILT reached 100 mm.

Part of the heat stored in the enclosure is lost to the outside world and the rest is released back into the greenhouse air. Effective heat storage was defined as the ratio of the total amount of heat released by the maintenance structure to the greenhouse air at night to the total amount of heat absorbed from the greenhouse air during the day. Figure 16 shows the comparison of the effective heat storage of different maintenance structures with different external insulation layers. When the external insulation layer of the NW and the SW was 20–80 mm, the effective heat storage ratio of the wall increased rapidly. It was proved that the thickening of the insulation layer had a very positive effect on the re-utilization of the heat stored in the wall. When EILT exceeded 80 mm, the effective heat storage ratio tended to be stable. The effective heat storage ratio of the SW was slightly higher than that of the NW, and the maximum heat storage ratio was 0.91. With the increase of EILT, the effective heat storage ratio, which was relatively

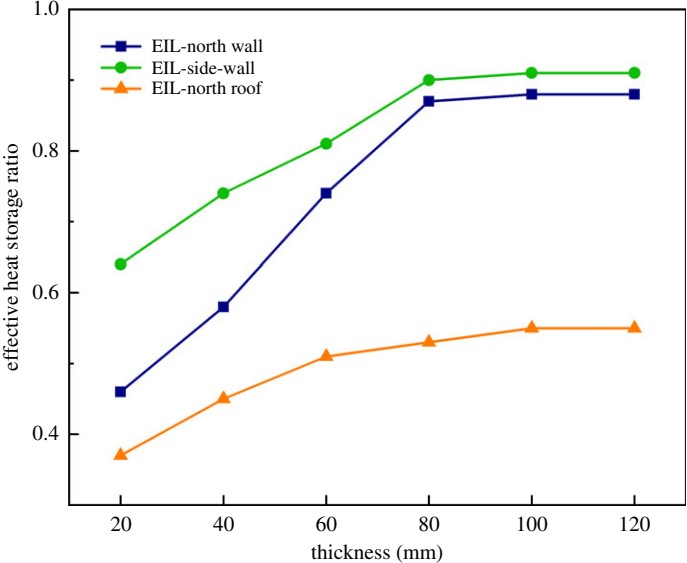

**Figure 16.** The comparison of the effective heat storage of different maintenance structures with different external insulation layers.

low on the NR, increased slowly. The maximum effective heat storage ratio was 0.55 when EILT was over 100 mm on the NR. It could be concluded that the EILT of the NW and SW could maintain the insulation effect of the maintenance structure at a relatively high level at 80 mm. The insulation layer on the NR should reach 100 mm. In general, compared with the NW and the NR, the SW could preferably release the heat absorbed from the air back into the air at night. The suitable thickness of the external insulation layer of each maintenance structure is obtained as follows: NW 80 mm, SW 80 mm, NR 100 mm.

# 4. Discussion

As can be seen from the results, the influence degree of the external thermal insulation layer on the greenhouse microclimate is as follows: SW > NW > NR. Covering the outer surface of the SW with insulation is the most cost-effective. Covering the insulation layer on the NW enhances the thermal performance significantly. When the budget is insufficient during the process of CSG construction, the first measure to take external thermal insulation coverage is the SW, followed by strengthening the external surface of the NW thermal insulation, which can effectively improve the heat storage and thermal insulation capacity. As the budget increases, the insulation of the SWs and the NR can be gradually enhanced. In the case of sufficient funds, the CSG thermal performance can be maximized by covering all the exterior surface of the enclosure with insulation layers. This study turns the empirical judgement of farmers' actual production into a theoretical basis for reference. Moreover, a scientific and comprehensive evaluation system has been proposed to determine the influence law of the external thermal insulation layer on the greenhouse microclimate environment. This model can be used to obtain the optimal external insulation configuration scheme with a limited cost budget and maximize the insulation performance of the greenhouse. The rational allocation of external insulation layer based on crop overwintering production in solar greenhouse in high-dimensional and cold areas is put forward, which reduces the waste of resources and strengthens the sustainable development of energy.

# 5. Conclusion

The present research is constructed by combining numerical simulation and experimental measurement with studying the effect of the external thermal insulation layer on the microclimate environment of CSG. The main obtained results are as follows:

(1) Covering the outer surface of the enclosures with thermal insulation could effectively increase the greenhouse temperature by 1.2–4.0°C. The unit costs of NW and CC were similar at USD 618.49 °C$^{-1}$ and USD 643.99 °C$^{-1}$, respectively. NR had the highest unit cost (USD 787.99 °C$^{-1}$), but it had the lowest increase in greenhouse air temperature. SW had the lowest unit cost (USD

282.52 °C$^{-1}$), but it limited the increase in greenhouse air temperature. The influence degree of the external thermal insulation layer on the greenhouse microclimate is as follows: SW > NW > NR.

(2) In high-dimensional and cold areas, CC as the suitable solution could raise the greenhouse air temperature maximally. The night-time heat release of the NW can reach 3666 KJ m$^{-2}$ with sufficient insulation thickness, while the SWs and the NR also reach 1590 and 850 KJ, respectively.

(3) The total heat released by the NW at night is significantly higher than that of the SW and the NR. The effective heat storage ratio of the SW was slightly higher than that of the NW, and the maximum heat storage ratio was 0.91. With the increase of EILT, the effective heat storage ratio, which was relatively low for the NR, increased slowly. The maximum effective heat storage ratio was 0.55 when EILT was over 100 mm on the NR. Compared with the NW and the NR, the SW can preferably release the heat absorbed from the air back into the air at night.

(4) In high-dimensional and cold areas, the suitable thickness of the external insulation layer of each maintenance structure is obtained as follows: NW 80 mm, SW 80 mm, NR 100 mm.

Ethics. This study does not involve ethical approval and informed consent.

Data accessibility. Effect of external thermal insulation layer on the Chinese solar greenhouse microclimate, Dryad, Dataset, https://doi.org/10.5061/dryad.x0k6djhjx.

Authors' contributions. Conceptualization: Z.F., X.L., L.Z., Y.L. Data curation: Z.F., X.L., Y.L., X.Y. Formal analysis: Z.F., X.L., Y.L., X.Y. Funding acquisition: Y.L. Investigation: Z.F., Y.L., X.Y., L.Z. Methodology: Y.L., X.Y., X.X. Project administration: X.X. Resources: X.Y., L.Z., T.L. Software: Z.F., X.L., Y.L. Supervision: Z.F. Validation: Z.F., Y.L., X.Y. Writing—original draft: Z.F. Writing—review and editing: Y.L. All authors gave final approval for publication and agreed to be held accountable for the work performed therein.

Competing interests. The authors declare that they have no known competing financial interests or personal relationships that could have appeared to influence the work reported in this paper.

Funding. This work was supported by The National Key Research and Development Programme of China (2019YFD1001905) and China Agriculture Research System of MOF and MARA.

Acknowledgements. We thank all members of National and Local Joint Engineering Research Center of Northern Horticultural Facilities Design and Application Technology (Liaoning) for their helps.

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
