## [Peer Review File · Royal Society Open Science]

Review History

RSOS-211217.R0 (Original submission)

Review form: Reviewer 1

Is the manuscript scientifically sound in its present form?

Yes

Are the interpretations and conclusions justified by the results?

Yes

Is the language acceptable?

Yes

Do you have any ethical concerns with this paper?

No

Have you any concerns about statistical analyses in this paper?

No

Recommendation?

Accept with minor revision (please list in comments)

Comments to the Author(s)

The paper entitled "Effect of external thermal insulation layer on the Chinese solar greenhouse microclimate" is well written and structured. The mechanism of the external thermal insulation layer that affecting the microclimate environment of CSG was clarified. It is globally a decent work on the external insulation for the envelope structure of the CSG. The results of this study can provide basic theoretical guidance for the rational configuration of the thermal insulation layer in CSG production practice. The work is novel but needs some issues resolved prior to acceptance. I therefore recommend minor revision of the paper before possible publication in the journal.

1. In the first paragraph of the introduction, the applied area of solar greenhouse in China should be referenced to improve the reliability.
2. What are the time nodes for the raising and covering of the insulation blanket in the model setting?
3. In Figure 1 (b), a wavy line appears under the North roof, which is very unattractive.
4. Fig. 13 the temperature scale is too fuzzy for clear reading.
5. The symbols in formulas 1-4 should be further interpreted for clear understanding.
6. The research results show that the response of thermal insulation layer placed on the side wall to the indoor microclimate is greater than that of the north wall and the north roof, but the actual building area of the side wall is much smaller than them. Explain it.
7. In the results section, there are too many direct descriptions of the research results. So, some summative conclusions should be added.

Review form: Reviewer 2

Is the manuscript scientifically sound in its present form?

No

Are the interpretations and conclusions justified by the results?

Yes

Is the language acceptable?

Yes

Do you have any ethical concerns with this paper?

No

Have you any concerns about statistical analyses in this paper?

No

Recommendation?

Major revision is needed (please make suggestions in comments)

Comments to the Author(s)

The work focused on Effect of external thermal insulation layer on the Chinese solar greenhouse microclimate. Unfortunately, this reviewer has concerns on the novelty and contributions presented in the paper. In order to help improve the paper quality, my suggestions and comments are shown below.

- 1) Summary should be Abstract: qualitative results with quantitative data are necessary to support the conclusion of the work. Be careful to use the word 'optimal thickness', as the optimal results can only be obtained after optimisation.
- 2) Research questions are too general and common. The conclusion is superficial.
- 3) Introduction: Scientific gaps are vague and not straightforward. The reviewer concerns the novelty and contribution of this study. Originality novelty of the work has not been clearly presented. In order to improve the readability and help readers to follow, it is better to firstly list the identified scientific gaps, before introducing the research framework. Furthermore, it is better to summarise the novelty and contribution at the end of introduction, with point-by-point items.
- 4) Methodology needs to be improved with more details.
- 5) Results: Provide in-depth analysis on each table and figure. Try to provide a summary first using 1-2 sentences, then support it by detailed data in the table or figure.
- 6) Nomenclature- better to re-order following the alphabetical order
- 7) Figure quality needs to be improved, including resolution, font size, etc.
- 8) Pay attention to the grammar and proof read by native speakers. For example, in Page 3, line 18, 'The research on external insulation of greenhouse is lack science.' 'is lack' should be 'is lack of'
- 9) There should be a point in line 46, page 3.
- 10) Pay attention to the format of each variable. The font style and size are strange.
- 11) One separate section is suggested to be added, to describe Research limitations, challenges and future prospects.
- 12) Generally, the conclusions as-written seem to only offer a summary of the results, rather than concluding the work. Point-by-point conclusion with quantitative data is necessary.

Overall, this manuscript is well-written and described. However, major revision is required.

Decision letter (RSOS-211217.R0)

Dear Dr Li

The Editors assigned to your paper RSOS-211217 "Effect of external thermal insulation layer on the Chinese solar greenhouse microclimate" have now received comments from reviewers and would like you to revise the paper in accordance with the reviewer comments and any comments from the Editors. Please note this decision does not guarantee eventual acceptance.

Please submit your revised manuscript and required files (see below) no later than 21 days from today's (ie 15-Oct-2021) date. Note: the ScholarOne system will 'lock' if submission of the revision

is attempted 21 or more days after the deadline. If you do not think you will be able to meet this deadline please contact the editorial office immediately.

on behalf of Dr Ramesh Rayudu (Associate Editor) and R. Kerry Rowe (Subject Editor)
openscience@royalsociety.org

Associate Editor Comments to Author (Dr Ramesh Rayudu):

Comments to the Author:

Please address the specific concerns from the reviewers as they are very valid and will improve the quality of your manuscript.

Reviewer comments to Author:

Reviewer: 1

Comments to the Author(s)

The paper entitled "Effect of external thermal insulation layer on the Chinese solar greenhouse microclimate" is well written and structured. The mechanism of the external thermal insulation layer that affecting the microclimate environment of CSG was clarified. It is globally a decent work on the external insulation for the envelope structure of the CSG. The results of this study can provide basic theoretical guidance for the rational configuration of the thermal insulation layer in CSG production practice. The work is novel but needs some issues resolved prior to acceptance. I therefore recommend minor revision of the paper before possible publication in the journal.

1. In the first paragraph of the introduction, the applied area of solar greenhouse in China should be referenced to improve the reliability.
2. What are the time nodes for the raising and covering of the insulation blanket in the model setting?
3. In Figure 1 (b), a wavy line appears under the North roof, which is very unattractive.
4. Fig. 13 the temperature scale is too fuzzy for clear reading.
5. The symbols in formulas 1-4 should be further interpreted for clear understanding.
6. The research results show that the response of thermal insulation layer placed on the side wall to the indoor microclimate is greater than that of the north wall and the north roof, but the actual building area of the side wall is much smaller than them. Explain it.
7. In the results section, there are too many direct descriptions of the research results. So, some summative conclusions should be added.

Reviewer: 2

Comments to the Author(s)

The work focused on Effect of external thermal insulation layer on the

Chinese solar greenhouse microclimate. Unfortunately, this reviewer has concerns on the novelty and contributions presented in the paper. In order to help improve the paper quality, my suggestions and comments are shown below.

- 1) Summary should be Abstract: qualitative results with quantitative data are necessary to support the conclusion of the work. Be careful to use the word 'optimal thickness', as the optimal results can only be obtained after optimisation.
- 2) Research questions are too general and common. The conclusion is superficial.
- 3) Introduction: Scientific gaps are vague and not straightforward. The reviewer concerns the novelty and contribution of this study. Originality novelty of the work has not been clearly presented. In order to improve the readability and help readers to follow, it is better to firstly list the identified scientific gaps, before introducing the research framework. Furthermore, it is better to summarise the novelty and contribution at the end of introduction, with point-by-point items.
- 4) Methodology needs to be improved with more details.
- 5) Results: Provide in-depth analysis on each table and figure. Try to provide a summary first using 1-2 sentences, then support it by detailed data in the table or figure.
- 6) Nomenclature- better to re-order following the alphabetical order
- 7) Figure quality needs to be improved, including resolution, font size, etc.
- 8) Pay attention to the grammar and proof read by native speakers. For example, in Page 3, line 18, 'The research on external insulation of greenhouse is lack science.' 'is lack' should be 'is lack of'
- 9) There should be a point in line 46, page 3.
- 10) Pay attention to the format of each variable. The font style and size are strange.
- 11) One separate section is suggested to be added, to describe Research limitations, challenges and future prospects.
- 12) Generally, the conclusions as-written seem to only offer a summary of the results, rather than concluding the work. Point-by-point conclusion with quantitative data is necessary.

Overall, this manuscript is well-written and described. However, major revision is required.

===PREPARING YOUR MANUSCRIPT===

===PREPARING YOUR REVISION IN SCHOLARONE===

Author's Response to Decision Letter for (RSOS-211217.R0)

See Appendix A.

Decision letter (RSOS-211217.R1)

Dear Dr Li,

It is a pleasure to accept your manuscript entitled "Effect of external thermal insulation layer on the Chinese solar greenhouse microclimate" in its current form for publication in Royal Society Open Science. The comments of the reviewer(s) who reviewed your manuscript are included at the foot of this letter.

Please see the Royal Society Publishing guidance on how you may share your accepted author manuscript at <https://royalsociety.org/journals/ethics-policies/media-embargo/>. After publication, some additional ways to effectively promote your article can also be found here

<https://royalsociety.org/blog/2020/07/promoting-your-latest-paper-and-tracking-your-results>.

on behalf of Dr Ramesh Rayudu (Associate Editor) and R. Kerry Rowe (Subject Editor)
openscience@royalsociety.org

Associate Editor Comments to Author (Dr Ramesh Rayudu):
Associate Editor
Comments to the Author:
Congratulations on your acceptance!

Appendix A

Dear Editors and Reviewers:

Thank you very much for giving us an opportunity to revise our manuscript. Those comments are all valuable and very helpful for revising and improving our paper, as well as the important guiding significance to our researches. We have carefully studied the relevant comments and have made major revision on the manuscript for your kind consideration for publication. Revised portions are marked in red in the manuscript. The main corrections in the present work and the responses to the reviewer's comments are as follows:

Responds to the reviewer #1's comments:

Points in Favor:

The paper entitled "Effect of external thermal insulation layer on the Chinese solar greenhouse microclimate" is well written and structured. The mechanism of the external thermal insulation layer that affecting the microclimate environment of CSG was clarified. It is globally a decent work on the external insulation for the envelope structure of the CSG. The results of this study can provide basic theoretical guidance for the rational configuration of the thermal insulation layer in CSG production practice. The work is novel but needs some issues resolved prior to acceptance. I therefore recommend minor revision of the paper before possible publication in the journal.

Response: Thanks very much for your kind work and consideration on publication of our paper. On behalf of my co-authors, we would like to express our great appreciation to editor and reviewers. The inaccurate problems and inappropriate statements have been completely corrected in the revised manuscript. Moreover, benefited from your very important suggestions and comments, the whole manuscript has been clearly organized to avoid confusion.

Points Detracting:

1. In the first paragraph of the introduction, the applied area of solar greenhouse in China should be referenced to improve the reliability.

Response:

Thank you very much for your kind work and valuable suggestions. The application

area of solar greenhouse in China should be illustrated with relevant literature. As suggested, the correction has been made in the revised version.

2. *What are the time nodes for the raising and covering of the insulation blanket in the model setting?*

Response:

The model used in this study is a mathematical-physical model based on computational fluid dynamics. And the time of uncovering the solar greenhouse insulation was set to 8:30 am and 16:00 pm during the model setup.

3. *In Figure 1 (b), a wavy line appears under the North roof, which is very unattractive.*

Response:

As your suggestion, Figure 1 (b) has been modified to improve its clarity as suggested by the reviewer.

4. *Fig. 13 the temperature scale is too fuzzy for clear reading.*

Response:

As suggested by the reviewer, Figure 13 has been enhanced for overall clarity to improve readability in the revised manuscript.

5. *The symbols in formulas 1-4 should be further interpreted for clear understanding.*

Response:

Thanks to the valuable suggestions. We have reordered and defined the professional symbols that appear throughout the manuscript in the symbol table and explained them below the equations in the manuscript.

6. *The research results show that the response of thermal insulation layer placed on the side wall to the indoor microclimate is greater than that of the north wall and the north roof, but the actual building area of the side wall is much smaller than them. Explain it.*

Response:

Due to the large volume of the north wall of the greenhouse, the high heat storage performance leads to a large amount of heat accumulation in the middle of the north wall body, forming a heat buffer area (i.e. transition layer). While the side walls are small in size, the heat is not favorably retained in the side walls. The extremely high indoor-outdoor temperature difference leads to higher heat dissipation per unit area than that of the north wall. Therefore, the side walls have a strong response to the

thermal insulation of the external insulation layer.

7. In the results section, there are too many direct descriptions of the research results. So, some summative conclusions should be added.

Response:

We have followed the reviewers' comments by adding a summary description of the specific work in each subsection and reducing the description of the resulting phenomena accordingly. The implications of the results as well as further mechanisms are analyzed.

Responds to the reviewer #2's comments:

Points in Favor:

The work focused on Effect of external thermal insulation layer on the Chinese solar greenhouse microclimate. Unfortunately, this reviewer has concerns on the novelty and contributions presented in the paper. In order to help improve the paper quality, my suggestions and comments are shown below.

Response: Thank you very much for giving us this opportunity to revise the manuscript to improve the quality of the paper. We apologize for not making the novelty and contribution of the paper clear, leading to some trouble for the reviewers in reading this article. We have substantially revised the manuscript point to point to improve the paper quality.

Points Detracting:

1. Summary should be Abstract: qualitative results with quantitative data are necessary to support the conclusion of the work. Be careful to use the word 'optimal thickness', as the optimal results can only be obtained after optimisation.

Response:

Thank you very much for timely correction. A series of qualitative results with quantitative data have been added to the summary section to support the conclusions of the work. Moreover, we replace the word 'optimum thickness' with a quantitative

thickness, which meets the requirements within a certain range (10mm).

2. Research questions are too general and common. The conclusion is superficial.

Response:

The manuscript is not well presented to reflect the contribution of the research work. So we have made extensive modifications to enable the reader to better understand the research content of this research.

Insulation is particularly important of CSG design and construction. Solar greenhouse external insulations can be divided into soft insulation and hard insulation. The soft insulation is to cover the south roof with thermal insulation layer (i.e. heat preservation quilt). Many studies have explained the effect of insulation quilt on the thermal performance of greenhouse. The hard insulation consists of three parts: the north wall, the side wall and the north roof. Nevertheless, a majority of studies are only focused on the thermal insulation of the north wall. The importance of insulation for side walls and the north roof was ignored. It is worth noting that all three should be studied in a systematic combination. For another, the configuration of external insulation is not considered from the economic point of view. There is no systematic, complete and reasonable size of external insulation. At present, people mainly rely on construction experience to build external insulation. This practice only meets the needs of greenhouse thermal insulation, but ignores the economy of thermal insulation construction, resulting in a great waste of resources. Moreover, some greenhouses even reduce the thickness of insulation layer for saving costs, resulting in the phenomenon of freezing damage to crops in severe winter. In this study, the effect of external insulation layers on the solar greenhouse is explained completely. The rational allocation of external insulation layer based on crop overwintering production in CSG is put forward. This methodology can effectively reduce the waste of resources and strengthen the sustainable development of energy.

3. Introduction: Scientific gaps are vague and not straightforward. The reviewer concerns the novelty and contribution of this study. Originality novelty of the work has not been clearly presented. In order to improve the readability and help readers to follow, it is better to firstly list the identified scientific gaps, before introducing the research framework. Furthermore, it is better to summarise the novelty and contribution at the end of introduction, with point-by-point items.

Response:

Thank you very much for the advice. Identifying the scientific gaps is very helpful for

the reader to understand the novelty of the research as well as the contribution. We have made significant corrections to list the scientific gaps in this paper. Moreover, we summarize the novelty and contribution at the end of introduction, with point-by-point items.

4. Methodology needs to be improved with more details.

Response:

As your suggestion, the methodology has been improved with more details in the revised manuscript.

5. Results: Provide in-depth analysis on each table and figure. Try to provide a summary first using 1-2 sentences, then support it by detailed data in the table or figure.

Response:

Thank you for the advice. We have analyzed each table and figure in depth, adding 1-2 sentences to summarize the findings before each section of the results analysis.

6. Nomenclature- better to re-order following the alphabetical order.

Response:

Thank you for raising this point. The previous Nomenclature has been cancelled. In order to understand the formula more clearly and intuitively, we have explained the professional symbols below the equations in the manuscript.

7. Figure quality needs to be improved, including resolution, font size, etc.

Response:

Thank you for the advice. We have improved the figure quality, including resolution, font size, etc.

8. Pay attention to the grammar and proof read by native speakers. For example, in Page 3, line 18, 'The research on external insulation of greenhouse is lack science.' 'is lack' should be 'is lack of'.

Response:

Thanks for your suggestion. We feel sorry for the grammar. We have carefully rechecked and revised the manuscript. Moreover, we have asked two colleagues who are skilled English language to help polish our article.

9. There should be a point in line 46, page 3.

Response:

As suggested, it has been modified in the revised draft.

10. Pay attention to the format of each variable. The font style and size are strange.

Response:

We are sorry for the change of variable symbols due to the upload version. We have modified the manuscript format and re-uploaded it.

11. One separate section is suggested to be added, to describe Research limitations, challenges and future prospects.

Response:

Thank you for the advice. In the revised manuscript, one separate section is added to describe research limitations, challenges and future prospects.

12. Generally, the conclusions as-written seem to only offer a summary of the results, rather than concluding the work. Point-by-point conclusion with quantitative data is necessary.

Response:

As suggested, the conclusions were transformed into point-by-point conclusion with quantitative data.

Special thanks to you for your good comments.

We appreciate for Editors/Reviewers' warm work earnestly, and hope that the correction will meet with approval.

Once again, thank you very much for your comments and suggestions.

Yours sincerely,

Yi-Ming Li

E-mail: liyiming@syau.edu.cn